# The Prognostic Impact of Retinoid X Receptor and Thyroid Hormone Receptor alpha in Unifocal vs. Multifocal/Multicentric Breast Cancer

**DOI:** 10.3390/ijms22020957

**Published:** 2021-01-19

**Authors:** Alaleh Zati Zehni, Falk Batz, Aurelia Vattai, Till Kaltofen, Svenja Schrader, Sven-Niclas Jacob, Jan-Niclas Mumm, Helene Hildegard Heidegger, Nina Ditsch, Sven Mahner, Udo Jeschke, Theresa Vilsmaier

**Affiliations:** 1Department of Obstetrics and Gynecology, University Hospital Munich, LMU, 80337 Munich, Germany; Alaleh.Zati@med.uni-muenchen.de (A.Z.Z.); falk.batz@med.uni-muenchen.de (F.B.); aurelia.vattai@med.uni-muenchen.de (A.V.); till.kaltofen@med.uni-muenchen.de (T.K.); svenja.schrader@med.uni-muenchen.de (S.S.); Helene.Heidegger@med.uni-muenchen.de (H.H.H.); Nina.Ditsch@uk-augsburg.de (N.D.); Sven.Mahner@med.uni-muenchen.de (S.M.); Theresa.Vilsmaier@med.uni-muenchen.de (T.V.); 2Department of General, Visceral, Transplant, Vascular and Thoracic Surgery, LMU, Marchioninistraße 15, 81377 Munich, Germany; Sven.Jacob@med.uni-muenchen.de; 3Department of Urology, LMU, Marchioninistraße 15, 81377 Munich, Germany; Janniclas.Mumm@med.uni-muenchen.de; 4Department of Obstetrics and Gynecology, University Hospital, 86156 Augsburg, Germany

**Keywords:** breast cancer, focality, retinoid X receptor, thyroid hormone receptor, steroid hormone receptor, prognosis, unifocal, multifocal, multicentric

## Abstract

The aim of this study was to assess the prognostic value of the steroid hormone receptor expression, counting the retinoid X receptor (RXR) and thyroid hormone receptors (THRs), on the two different breast cancer (BC) entities: multifocal/multicentric versus unifocal. The overall and disease-free survival were considered as the prognosis determining aspects and analyzed by uni- and multi-variate analysis. Furthermore, histopathological grading and TNM staging (T = tumor size, N = lymph node involvement, M = distant metastasis) were examined in relation to RXR and THRs expression. A retrospective statistical analysis was carried out on survival-related events in a series of 319 sporadic BC patients treated at the Department of Gynecology and Obstetrics at the Ludwig-Maximillian’s University in Munich between 2000 and 2002. The expression of RXR and THRs, including its two major isoforms THRα1 and THRα2, was analyzed by immunohistochemistry and showed to have a significant correlation for both BC entities in regard to survival analysis. Patients with multifocal/multicentric BC were exposed to a significantly worse disease-free survival (DFS) when expressing RXR. Patients with unifocal BC showed a significantly worse DFS when expressing THRα1. In contrast, a statistically significant positive association between THRα2 expression and enhanced DFS in multifocal/multicentric BC was shown. Especially the RXR expression in multifocal/multicentric BC was found to play a remarkably contradictory role for BC prognosis. The findings imply the need for a critical review of possible molecular therapies targeting steroid hormone receptors in BC treatment. Our results strengthen the need to further investigate the behavior of the nuclear receptor family, especially in relation to BC focality.

## 1. Introduction

Breast cancer (BC) is the most frequent malignant tumor in women worldwide [1]. With 2.1 million incident cases in 2018 [2,3], half a million deaths, and 14.9 million disability-adjusted life-years [4], breast cancer is considered one of the greatest challenges for experts to control [5]. Options for treatment of breast cancer have advanced greatly over the past years. The therapy intention is either of a curative nature or for the purpose of survival prolongation in metastatic BC and thus for the preservation of quality of life. Therapy regimes in adjuvant, neoadjuvant, and metastatic settings are immensely reliant on clinical tumor subtypes and include chemotherapy, surgery, aromatase inhibitors and hormone-receptor modulators [6,7,8].

For distinguishing clinical tumor subtypes, the BC focality is recognized as an important prognostic factor. The definition of multifocal BC states two or more separate tumor loci in the same quadrant. Multicentric BC is understood as two or more separate invasive tumors in more than one quadrant of the same breast [9,10]. To this point, no standard international definition has been implemented for distinguishing multifocal and multicentric BC [11,12]. Consequently, in our study, the multifocal and multicentric BC patients have been merged into one multifocal group to allow a distinct comparison with the unifocal BC patient group. The focality is a significant factor affecting the progressive course of disease and has been described by multiple studies for multifocal and multicentric BC patients [13]. Multifocality and/or multicentricity was found to be predictive of a worse prognosis through increased rates of distant metastasis, local relapse and shorter survival [14], prevalence of lymph node metastases [15], and higher mortality rates [16]. In contrast, unifocal BC is associated with an enhanced prognosis, including a better overall survival (OS) and disease-free survival (DFS), in comparison to multifocal and/or multicentric BC with the same tumor size. The focality is consequently regarded as an important prognosticator for BC [14,15,16].

The estrogen receptor (ER) and progesterone receptor (PR) are established “classical steroid hormone receptors” and are a key decision component in therapeutic approaches and forecasting prognosis for BC patients. There is strong evidence that the involvement of other nuclear receptors, besides ER and PR, play a vital role in breast cancer biology, including development and progression [17]. Personalized treatment options nowadays already involve drugs that target nuclear receptors [18]. The retinoid X receptor (RXR), thyroid hormone receptors (THRs), and Vitamin D receptor (VDR), are all members of the nuclear hormone receptor superfamily [19] and are ligand-dependent transcription factors, which bind several lipophilic hormones and lipid metabolites [20]. Our previous study has already identified the VDR in multifocal BC as an independent prognostic marker for a worsened OS. Interestingly, for the unifocal breast tumor patients, the VDR showed a significant positive association regarding the course of disease [21]. THRs have been identified to assemble with VDR and RXR by forming functional heterodimers. However, so far neither RXR nor THRs have been studied in association to breast cancer focality.

RXR and THRs activation is achieved by binding with its ligands and has been identified to form homodimers and heterodimers with many other members of the nuclear receptor superfamily [22]. After forming heterodimers [23], they (a) translocate to the nucleus, (b) bind to specific response elements upon promoters of specific genes, and (c) act as transcription factors [24]. Diverse ligands bound to these receptors, on the other hand, recruit different co-activators and consequently regulate different genes and biological functions [24].

Retinoids derived from vitamin A are signaling molecules that act via RXRs and are key components in cell differentiation and proliferation [25]. Retinoids have been previously described for their ability to induce differentiation and arrest proliferation in cancer cells [20,25]. To this point, three subtypes of the RXR have been identified: RXRα (NR2B1), RXRβ (NR2B2), and RXRγ (NR2B3) [26]. RXR is described to be expressed by breast cancer cells [27]. Increased expression of RXRα was identified in breast cancer cells rather than benign breast tissue [28]. These receptors are documented to have tumor suppressor properties in mediating the anti-proliferative effects of retinoic acid and inhibiting cell proliferation [25]. Evidence states that activation of RXR induces apoptosis in breast cancer cells and may reduce cell growth [29] in vitro and in animal models [30,31], also in combination with selective ER modulators [32,33]. Several studies, including Heublein et al., suggest that RXR positivity may predict favorable prognosis in breast cancer and comprise anti-cancer cell activity [34,35,36]. Overall, these findings suggest that RXR plays a key function in tumor pathogenesis.

Evidence implies a correlation between BC and thyroid disorders. Patients with thyroid dysfunctions show increased breast cancer incidences in contrast to healthy women [37,38]. Ditsch et al. [39] observed increased blood levels of the thyroid hormones (TH) fT_3_ and fT_4_ and concentrations of thyroid stimulating hormone (TSH) and antibodies against thyroidal peroxidase at the time of primary diagnosis in BC patients [40]. Circulating THs bind to THRs, endorse downstream signaling pathways and activate transcription factors [41]. Four major THRs isoforms have been identified: THRα1, THRα2, THRβ1, and THRβ2 [42]. THRα1, THRα2 and THRβ1 are overexpressed in several tissues in the human body except the liver as the major TH target organ [42,43]. Transcriptional activity is regulated by THRs through homo- or hetero-dimers with other nuclear receptors such as RXR and VDR. RXR forms a heterodimer with THRs and together influence downstream target gene expression by binding to specific DNA sequences that are located in regulatory regions identified as thyroid hormone elements [41,44,45]. THRs have been identified to be highly expressed in breast cancer tissue deriving from patients diagnosed with a *BRCA1* germline mutation [34]. Further, THRs were of opposing prognostic significance and silencing of THRα appeared to diminish viability of *BRCA1* mutated BC cells [46].

No study so far has identified the expression of RXR and THRs in human breast cancer specimens in regard to focality. New insights could potentially be promising in regard to cancer therapeutics. The above-mentioned nuclear receptors could consequently provide supplementary therapeutic targets for breast cancer patients.

Being aware of the immense development of todays clinical oncology towards customized treatment options, this study focused on investigating steroid hormone receptor expression in unifocal versus multifocal/multicentric sporadic BC and its influence on recurrence and survival. THRs and RXR, both nuclear receptors and activated by their steroid hormones, could be significant targets for generating new therapy treatments and prevention of BC. This study aims to provide a scientific base for future BC endocrine therapies adjusted to focality type, with the intetion to excert effectivness and decrease toxic treatment.

## 2. Results

### 2.1. Retinoid X Receptor (RXR)

#### 2.1.1. Unifocal BC

Investigating the association of RXR expression on BC prognosis, no statistically significant difference was observed, neither for the OS (*p* = 0.360), nor for the DFS (*p* = 0.942), calculated by the log rank test. In addition, all three categories of TNM Staging (pT *p* = 0.440, pN *p* = 0.068, pM *p* = 0.673) and the histopathological grading by WHO (*p* = 0.738) revealed no significant difference between RXR positive or negative patients.

#### 2.1.2. Multifocal and/or Multicentric BC

In multifocal and/or multicentric BC RXR expression showed no significant effect on the OS (*p* = 0.521). Yet, the Kaplan–Meier curve visualized (Figure 1) and the log rank test calculated with a *p* value of 0.036 show a significant negative association of the DFS in multifocal/multicentric BC patients when expressing the RXR. Interestingly the histopathological TNM staging (pT *p* = 0.328, pN = 0.820, pM = 0.497) and tumor grading by WHO (*p* = 0.466) Kruskal–Wallis analysis revealed no statistical difference for RXR expression in multifocal and/or multicentric BC. When conducting multivariate Cox regression, the RXR was identified as a dependent prognostic factor in the unifocal group for the DFS (HR 1.547, 95%CI 0.87–3.483, *p* = 0.292) (Table 1).

### 2.2. Thyroid Hormone Receptor α1 (THRα1)

#### 2.2.1. Unifocal BC

The THRα1 was the only receptor in this study, showing a significant effect on unifocal BC. The Kaplan–Meier curve illustrated a worse DFS for unifocal BC patients, when expressing the THRα1. This finding was confirmed by the Log-Rank test with a highly significant *p* value of 0.012 (Figure 2). Regarding the OS of unifocal BC patients, THRα1 expression revealed no statistically significant difference (*p* = 0.524). Additionally, no significant relation between THRα1 expression and TNM staging and WHO grading was calculated by Kruskal–Wallis analysis (pT p= 0.469; pN *p* = 0.464; pM p= 0.076; grading *p* = 0.470). Multivariate Cox regression did not identify the THRα1 as an independent prognostic factor for the DFS (HR 1.626, 95%CI 0.978–1.022, *p* = 0.721) (Table 2).

#### 2.2.2. Multifocal and/or Multicentric BC

Using the same statistical devices for the multifocal and/or multicentric BC group, no significant correlations between prognosis and THRα1 expression could be outlined (DFS *p* = 0.617; OS *p* = 0.564). Likewise, the Kruskal–Wallis tests revealed no significant results for tumor size pT (*p* = 0.479), involution of local lymph nodes (*p* = 0.255), the presence or lack of metastases (*p* = 0.494) or histopathological tumor grade (*p* = 0.325) at initial diagnosis for the multifocal and/or multicentric group.

### 2.3. Thyroid Hormone Receptor α2 (THRα2)

#### 2.3.1. Unifocal BC

Unifocal BC patients revealed no significant correlations between THRα2 expression and prognosis in this study. Neither the OS (*p* = 0.199), nor the DFS (*p* = 0.243) were significantly affected by the THRα2, calculated by Log-Rank test. In line with these results, this receptor showed significant effects when tested for grading (*p* = 0.079) and staging (pT *p* = 0.699, pN *p* = 0.491, pM *p* = 0.180), calculated with the already mentioned statistical devices.

#### 2.3.2. Multifocal and/or Multicentric BC

Like all the other analyzed receptors and cohorts, the OS was not affected by the lack or existence of the THRα2 (*p* = 0.053). A highly significant *p* value of 0.000 was calculated for the DFS by Log-Rank test. The Kaplan–Meier survival analysis visualized that patients with multifocal and/or multicentric BC have a better DFS, when expressing the THRα2 (Table 3). TNM Staging (pT *p* = 0.869, pN *p* = 0.069, pM *p* = 0.561) and WHO grading (*p* = 0.648) was, as well as all the receptors listed above, were not significant for the THRα2 in this cohort. In addition, the THRα2 was not an independent prognosticator for the DFS, when conducting multivariate COX regression (HR 0.742, 95% CI 0.370–1.486, *p* = 0.399) in the multifocal and/or multicentric group (Figure 3).

## 3. Discussion

The aim of this study was to evaluate the prognostic association of the steroid hormone receptor RXR and THRs expression in relation to the two different BC entities: unifocal vs. multifocal BC. Understanding the mechanisms by which RXRs and THRs exert their effects in breast cancer patients remains incomplete [25]. This is the first study to define the prognostic role of THRs and RXR in breast cancer in relation to the two different BC entities, using a relatively large clinical patient cohort with long-term follow up. Results from the current study provide evidence that expression of RXR and THRs showed a significant association in terms of the course of BC disease in relation to focality.

RXR and THRs, both nuclear receptors with their associated ligands, operate as potent regulators of cell differentiation, development, and normal physiology. Furthermore, they might play an important role in different pathologies including breast cancer [22,47]. As previously described, the RXR, THR, and VDR have been identified to form functional homodimers and heterodimers with many other members of the nuclear receptor superfamily, also in human breast cancer cell lines [22]. These were able to mediate selective responses such as growth inhibition and apoptosis, supporting initially a protective role for breast cancer development [30]. As these molecules are considered as potential targets for molecular therapy [29,30], the aim of these studies was to evaluate the RXR and TRHs expression in tissue of BC patients and to correlate it with major clinicopathological characteristics and prognostic factors, in relation to BC focality. Evaluation of the immunoreactive score of Remmele and Stegner (IRS) was performed as both RXR and TRH belong to the nuclear receptors family for which the IRS is commonly used [48].

The nuclear receptor RXR has been proven to modulate cellular differentiation and apoptosis in different tumor entities [49]. RXR and its heterodimers regulate the function of myeloid cells, link cellular metabolism, and show a profound effect on immune function [50,51]. Previous studies have concluded that the specific activation of RXR may up regulate chemokine expression and promote phagocytosis of apoptotic cells. Further, it can decrease antiviral responses in myeloid cells [50,52]. Increased expression of RXR has been related with an up-regulated apoptosis in ovarian cancer and a specific RXR activation in an ovarian tumor-model even showed an apoptosis re-activation [53]. In lung cancer, the epigenetic silencing of RXR was associated with decreased OS while the exact mechanism for this result is still unknown [54]. In in vitro studies on BC, RXR ligands or retinoids are reported to induce apoptosis in BCL2-positive human cancer cells [29], and decreased vascularization in BC tumors in transgenic mice [55]. Furthermore, RXR agonist therapy also suppressed mammary tumorigenesis in transgenic mice [56]. Additionally, RXR activation down regulated the COX-2 expression in BC cells [57], and blocked the BC cell cycle at the G1 phase [58]. Nevertheless, in a clinical trial for treating metastatic BC, a retinoid agonist was ineffective [59]. On the other hand, *BRCA1* mutated breast cancer cells have been hypothesized to be sensitive to RXR and VDR modulating drugs [34].

Interestingly, the results of our study do not support the tumor-inhibiting role of RXR. Patients with multifocal/multicentric BC showed a significantly worse DFS when expressing RXR. In contrast, no significant correlation between RXR expression was noted in survival analysis for unifocal BC. In addition, no correlation between RXR expression and TNM staging or grading was found. In line with our previous findings, also the VDR expression showed to play a remarkably paradox role for BC prognosis. The multifocal/multicentric BC patients with significantly worse DFS revealed enhanced expression levels of the VDR. We even identified the VDR to be an independent prognostic marker for multifocal BC patients [21]. We propose that increased expression of RXR might potentiate heterodimer formation and activation of other nuclear receptors such as VDR, thus increasing a possible tumorigenic function in multifocal BC. We therefore suggest a fundamental interaction between RXR and VDR and their heterodimers in multifocal BC patients.

Despite previous histological data supporting an anti-tumorigenic effect of elevated RXR and VDR expression, the findings of our study group seem to rather support the opposite in multifocal BC. Since the exact mechanism is still unclear, these data strengthen the need to further investigate the behavior of the nuclear receptor family including RXR and VDR in BC, especially in relation to focality. Due to the application of the IRS, our findings should be considered with great care. If, however, this holds true if proven by larger series, we suggest a critical reconsideration of possible RXR and Vitamin D therapy approaches subjected to down-regulation along the BC progression and continue further research of the steroid hormonal receptor pathogenesis in BC subtypes.

Likewise, THRs form homodimers or heterodimers with RXR and are later activated by thyroid hormones. Consequently, they act as classical transcription factors by binding to the promoter regions of target genes [60]. THRs are encoded by two genes: THRα and THRβ—on chromosome 17 and 3, respectively [60,61,62]. To this date, there is still little knowledge regarding the specific THRα isoforms: THRα1 and THRα2. As described previously, THRα is expressed in various organ tissues and its analogous malignant tissues, yet its clinical relevance and role in BC etiology and progression remains unclear [63,64,65,66]. The functional similarity of THRs and ER/PR have previously led to the hypothesis that THRs may be a prognostic marker in breast cancer patients [67]. Ditsch et al. revealed an association between lower THRα2 expression and worse survival outcome in general BC patients [68]. Another study by Conde et al. exposed a significant correlation between high THRα expression and DFS in BC patients, however, without assessing the specific THRα isoforms individually [64].

In our study, THRα1 and THRα2 showed a prognostic association between both BC entities, but with major differences. Patients with unifocal BC showed a significantly worse DFS when expressing THRα1. In contrast, no significant correlation between THRα1 expression was noted in survival analysis for multifocal/multicentric BC. On the other hand, there was a statistically significant positive association between high THRα2 expression and enhanced DFS in multifocal/multicentric BC. No statistically significant association was found for unifocal BC and THRα2 expression. There was no significant correlation of the THRα isoforms concerning TNM, histopathological grading, and staging. Our findings are congruent with previous outcomes regarding THRα expression and their effect on survival analysis in BC. Similarly, Jerzak et al. described low THRα1 expression and high THRα2 expression in general BC patients that had the highest observed 5-year OS [63]. Nevertheless, our study, for the first time, identified THRα expression in human breast cancer specimens concerning focality.

Opposing molecular pathways of the THRα isoforms may explain the cause for differing effects on survival analysis. Whilst THRα1 is activated by thyroid hormone [69], THRα2 lacks the binding site for thyroid hormone [63,70]. In detail, THRα2 serves as an antagonist of thyroid hormone-mediated biological effects and signaling, preventing over- or under-activity of thyroid-resolved effects. Based on our results, expression of THRα2 may antagonize signaling of thyroid hormone growth-promoting effects that are mediated by THRα1 [42,70,71]. While the mechanism underlying this finding has not been determined, it is possible that unifocal BC patients with predominant THRα1 expression may benefit from reducing thyroid hormone concentrations and/or inhibiting THRα1 [72,73]. It is hypothesized that THRα2 expression reduces growth-promoting genes in breast cancer by decreased transcription of p53 and retinoblastoma [74]. We hypothesize that multifocal/multicentric BC patients may profit from the up-regulation of THRα2. Especially for these BC focality entities, with an unfavorable prognosis; the survival outcomes could be improved [75].

Distinguishing the BC entities may be regarded as the most important limitation of this study. Dividing BC entities into its subtypes may differ depending on pathological centers and examiners. Especially for multifocal BC, defined by two or more separate tumor loci in the same quadrant, minimal distance between the separate tumors may result to be considered as unifocal BC. Thus, distinction between unifocal and multifocal BC may not always be clear. Additionally, to date, no standard international definition has been implemented for distinguishing multifocal and multicentric BC [11,12]. So far, several studies including Weissenbacher et al. have hypothesized that the two entities were found to be predictive of a worse prognosis. The question of whether multifocal and multicentric BC can be regarded as equivalent in terms of aggressiveness of the disease should become the subject of further investigations.

## 4. Materials and Methods

### 4.1. Patients

The cohort for this study is built of patients with BC treated in the years of 2000 to 2002 at the Department of Gynecology and Obstetrics at the Ludwig-Maximillian’s University in Munich.

The study’s aim was to assess the prognostic value of steroid hormone receptors on the two different BC entities: unifocal vs. multifocal and/or multicentric sporadic BC. According to the current state of research, we decided to include and further investigate the most relevant steroid hormone receptors: RXR and THRs and here both alpha isoforms, counting THRα1 and THRα2. As described in Section 4.2, selected samples were immunohistochemically stained (Figure 4) and statistically analyzed in the manner described Section 4.3.

To determine the focality, the recruits had to undergo set clinical diagnostics: clinical examination, ultrasound and X-ray. If the identification of the focality still was indistinct, further diagnostics such as nuclear magnetic resonance imaging (NMRI), pneumocystography or galactography, were added. Finally, we excluded patients with an unclear receptor status and/or focality type from our database. In the end, 319 patients all meeting the set requirements built our total collective (TC) (Table 4). With the TC database, survival analysis was performed for each receptor and always in regard to the focality. After an observation period of up to 10 years DFS and OS were statistically analyzed, this follow-up data were retrieved from the Munich Cancer Registry.

For further investigation, e.g., for the TNM staging [76,77], WHO grading [78] and multivariate analysis, the TC was subdivided into two groups, depending on the focality. Group 1 contained all patients with unifocal BC with a total of 173 patients. To make it clearer and more comprehensible, we merged the multifocal and multicentric BC cases to Group 2, including 146 patients.

In Table 4, detailed patient characteristics from the TC are summarized and displayed. The large TC of 319 patients and the relatively equal distribution in the subgroups strengthen the statistical power of our study. The median age at initial diagnosis from the 319 patients included was 59 years with a range of 69. Overall, 173 patients were diagnosed with unifocal and 146 with multifocal and/or multicentric BC; 61.4% of the patients had histological invasive carcinoma of no special type (NST); 52.2% of our patients had a low-grade carcinoma (G1-G2 = 52.2%, G3 = 47.7%) and 64.3% were staged with a tumor size smaller than 2 cm (pT1: 64.3%, pT2-pT4: 35.6); 54.2% of all patients were staged pN0. The majority of our TC was staged with no present metastasis at initial diagnosis (pM0 = 78.1%, pM1 = 21.8%). The negligibly different total patient numbers (N) in the subgroups may be explained by the lack of a limited number of input variables that could not be obtained by the retrospective character of the study.

### 4.2. Immunohistochemistry

According to the earlier published and well described methods [46,68,79], immunohistochemistry of RXR and THRα on formalin-fixed, paraffin-embedded sections was performed. Therefore, a combination of pressure cooker heating and the standard streptavidin–biotin–peroxidase complex with mouse/rabbit-IgG-Vectastain Elite ABC kit (Vector Laboratories, Burlingame, CA, USA) were used. For the staining, we utlizied the following anibodies: the THRα1 was stained with polyclonal rabbit IgG antibodies (AbD Serotec Oxford, UK), the THRα2 by using monoclonal rabbit IgG1 (AbD Serotec, Oxford, UK) Abcam, Cambridge, UK); Zytomed, Berlin, Germany) and the RXR was detected by monoclonal mouse antibodies (Perseus Proteomics Inc., Tokyo, Japan). For the negative controls, instead of the primary antibody we used appropriate tissue sections, which were treated with pre-immune IgGs (supersensitive rabbit negative control, BioGenex, Fremont, CA, USA). Figure 4 contains an exemplary presentation of immunohistochemical stained steroid receptors. Positive-stained tissue appeared in a brownish color (Figure 4A,C) and negative as well as unstained cells appeared in blue (Figure 4B,D).

To quantify the immunoreactivity, meaning the distribution and intensity patterns, two blinded and independent observers evaluated via semi-quantitative immunoreactive score of Remmele and Stegner (IRS) [48] by using a Leitz microscope (Wetzlar, Germany) and a 3CCD color camera (JVC, Victor company of Japan, Yokohama, Japan). The IRS scoring system ranges from 0 to 12. Therefore, the staining intensity (Score 0 = no staining, Score 1 = weak staining, Score 2 = moderate staining, Score 3 = strong staining) needed to be multiplied with the percentage of positively stained cells.

Tissue samples that had been assigned an IRS greater than 3 for the RXR and THR α1 were scored as positive. For the THR α2 we assessed an IRS higher than 1 to be positive.

### 4.3. Statistical Analysis

In this study, statistical analysis was performed by using the IBM Statistical Package for the Social Sciences (IBM SPSS Statistic 24.0 Inc., Chicago, IL, USA). The collected results were inserted into the SPSS database in the implied manner, building the TC.

Using the TC database, we analyzed the effect of the initially defined three receptors on the OS and DFS always with regard to the focality. By applying the chi-square of the log rank test, we tested for significance. Kaplan–Meier survival analysis was performed for the visualization of each steroid receptor. For the statistical evaluation of the TNM staging and histopathological WHO grading, we divided the TC database into two subgroups due to the BC focality: Database 1 including all patients with unifocal BC and Database 2 containing all patients with multifocal and/or multicentric BC. Here, the nonparametric Kruskal–Wallis test for significance and boxplots were used to examine variables. Multivariate analyses via Cox regression evaluated the dependency as a prognostic marker of each receptor, when adjusted for age, staging and grading. Each parameter to be considered significant in our study was required to have a *p* value less than 0.05.

## 5. Conclusions

In conclusion, the present study analyzed the prognostic association of the steroid hormone receptors’ expression of RXR and THRα in relation to the different BC entities: unifocal vs. multifocal/multicentric BC. The RXR expression was shown to play a remarkably incongruous role for BC prognosis in comparison to previous findings. Patients with multifocal/multicentric BC were exposed to a significantly worse DFS when expressing RXR. In line with our previous findings, also the VDR expression previously showed to play a remarkably contradictory role for BC prognosis [21]. Despite previous reports supporting an anti-tumorigenic effect of elevated RXR and VDR expression, our results seem to rather support the opposite effect for multifocal BC patients. The findings imply a critical review of possible molecular therapies involving RXR and Vitamin D as a target in BC treatment. If proven by larger series, future therapy decisions should be made in hindsight of BC focality. THRα1 and THRα2 showed a prognostic association in both BC entities, but with major differences. Patients with unifocal BC showed a significantly worse DFS when expressing THRα1. In contrast, a statistically significant positive association between THRα2 expression and enhanced DFS in multifocal/multicentric BC was shown. Our findings are congruent with previous outcomes regarding THRα expression and its effect on survival analysis in breast cancer. Our study, for the first time, identified THRα expression in human breast cancer specimens in regard to focality. In summary, thyroid hormone-modulated therapies should become the subject of further investigations in regard to BC focality. Our results strengthen the need to further investigate the behavior of the nuclear receptor family in BC, especially in relation to focality. Further examinations studying the cause and to what extent BC focality may impact hormonal effects would be of major interest.

## Figures and Tables

**Figure 1 ijms-22-00957-f001:**
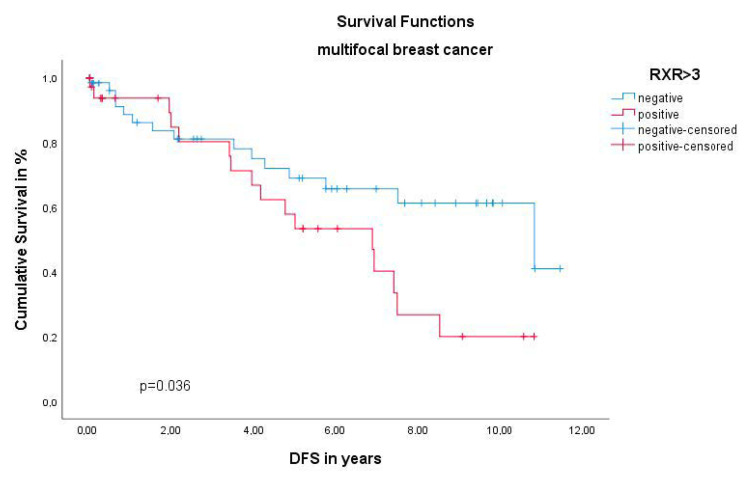
Kaplan–Meier survival analysis among retinoid X receptor (RXR) positive and negative patients. Disease-free survival (DFS) of patients with multifocal and/or multicentric breast cancer (BC).

**Figure 2 ijms-22-00957-f002:**
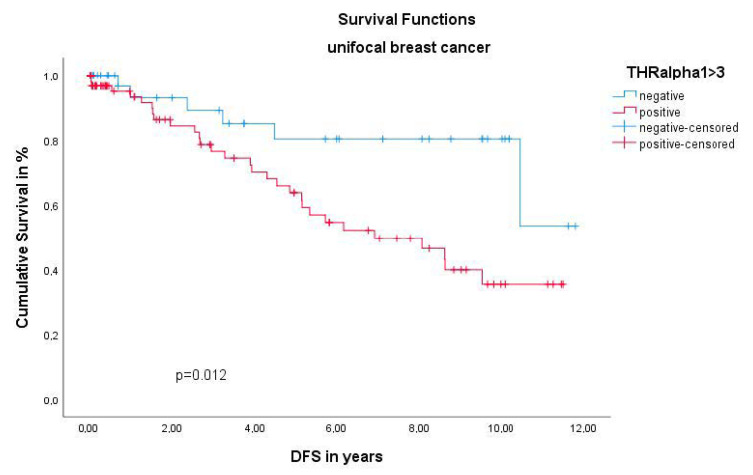
Kaplan–Meier survival analysis among thyroid hormone receptor α1 (THRα1) positive and negative patients. DFS of patients with unifocal BC.

**Figure 3 ijms-22-00957-f003:**
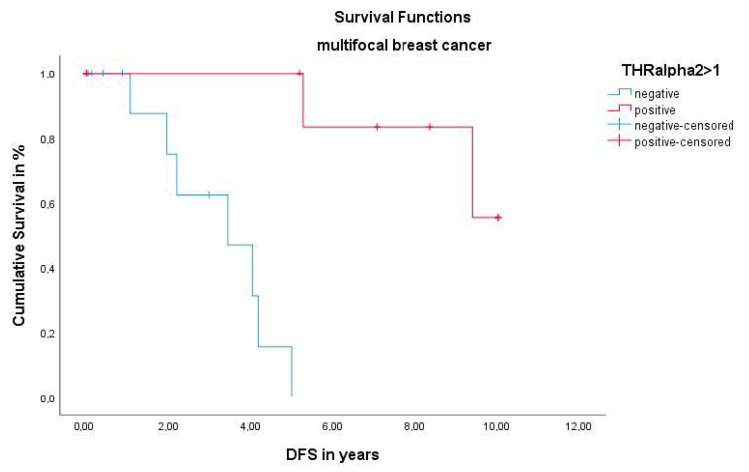
Kaplan–Meier survival analysis among THRα2 positive and negative patients. DFS of patients with multifocal and/or multicentric BC.

**Figure 4 ijms-22-00957-f004:**
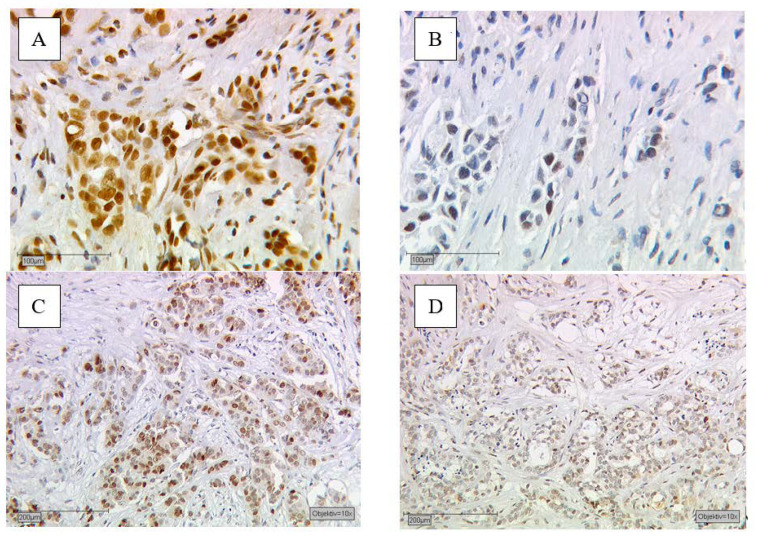
Shows immunohistochemical staining of RXR and THRα2 after incubation with the primary antibody of the malignant breast cancer cells. Immunohistochemical staining of RXR in human BC (**A**,**B**). (**A**) With an immunoreactive score of Remmele and Stegner (IRS) of 8 meaning RXR positive and (**B**) an IRS value ≤ 2 being RXR negative. Immunohistochemical staining of THRα2 in BC (**C**,**D**). (**C**) With an IRS of 9 meaning THRα2 positive and (**D**) an IRS value ≤ 1 being THRα2 negative.

**Table 1 ijms-22-00957-t001:** Multivariate Cox regression analysis of multifocal and/or multicentric BC patients regarding DFS.

Variable.	Coefficient	HR (95% CI)	*p* Value
**Age**	−0.001	0.999 (0.970–1.0301)	0.967
**Grading**	−0.008	0.992 (0.985–0.999)	**0.034**
**pT**	0.215	1.240 (0.833–1.845)	0.290
**pN**	−0.003	0.997 (0.982–1.012)	0.695
**pM**	2.353	10.516 (4.694–23.559)	**0.000**
**RXR**	0.437	1.547 (0.687–3.483)	0.292

Significant results are shown in bold; HR: hazard ratio; CI: confidence interval.

**Table 2 ijms-22-00957-t002:** Multivariate Cox regression analysis of unifocal BC patients regarding DFS.

Variable	Coefficient	HR (95% CI)	*p* Value
**Age**	0.010	1.010 (0.975–1.046)	0.575
**Grading**	0.418	1.159 (0.761–3.033)	0.236
**pT**	0.115	1.122 (0.784–1.605)	0.530
**pN**	0.022	1.022 (0.795–1.314)	0.867
**pM**	2.079	7.993 (3.007–21.248)	**0.000**
**THRα1**	0.486	1.626 (0.532–4.973)	0.394

Significant results are shown in bold; HR: hazard ratio; CI: confidence interval.

**Table 3 ijms-22-00957-t003:** Multivariate Cox regression analysis of multifocal and/or multicentric BC patients regarding DFS.

Variable	Coefficient	HR (95% CI)	*p* Value
**Age**	0.005	1.005 (0.978–1.033)	0.721
**Grading**	−0.008	0.984 (0.984–0.999)	**0.033**
**pT**	0.201	1.222 (0.841–1.776)	0.293
**pN**	−0.001	0.999 (0.984–1.014)	0.880
**pM**	2.550	12.812 (5.662–28.988)	**0.000**
**THRα2**	−0.299	0.742 (0.370–1.486)	0.399

Significant results are shown in bold; HR: hazard ratio; CI: confidence interval.

**Table 4 ijms-22-00957-t004:** Patient characteristics of the total collective.

Patient Characteristics	n (%)
Age (years)	Median 59.09 Range 69
Tumor foci	Unifocal 173 (54.2) Multifocal 146 (45.7)
Histology	NST 188 (61.4) Non-NST 118 (38.5)
Tumor grade	G1 or G2 165 (52.2) G3 151 (47.7)
pT	pT1 197 (64.3) pT2-pT4 109 (35.6)
pN	pN0 166 (54.2) pN1-pN3 140 (45.7)
pM	pM0 239 (78.1) pM1 67 (21.8)
RXR	negative 186 (58.3) positive 133 (41.6)
THRα1	negative 120 (37.6) positive 199 (62.3)
THRα2	negative 172 (53.9) positive 147 (46.0)

## Data Availability

Data are accessible on request from the authors.

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
