# Peer review of "The Prognostic Impact of Retinoid X Receptor and Thyroid Hormone Receptor alpha in Unifocal vs. Multifocal/Multicentric Breast Cancer"

_ijms, 2021, doi:10.3390/ijms22020957_

Round 1

Reviewer 1 Report

This is a nicely written manuscript with interesting findings. The study assesses the prognostic value of the steroid hormone receptor expression, counting the retinoid X receptor (RXR) and thyroid hormone receptors (THRs), on the two different breast cancer (BC) entities: multifocal/multicentric versus unifocal. The endpoints considered were overall- and disease-free survival, with two major isoforms THRα1 and THRα2 showing correlation with survival analysis. The study would gain more strength if it could present the findings by summarizing the challenges and opportunities and provide a framework for future directions, but this may be beyond the scope of the current study.

Author Response

Comments and Suggestions for Authors

This is a nicely written manuscript with interesting findings. The study assesses the prognostic value of the steroid hormone receptor expression, counting the retinoid X receptor (RXR) and thyroid hormone receptors (THRs), on the two different breast cancer (BC) entities: multifocal/multicentric versus unifocal. The endpoints considered were overall- and disease-free survival, with two major isoforms THRα1 and THRα2 showing correlation with survival analysis. The study would gain more strength if it could present the findings by summarizing the challenges and opportunities and provide a framework for future directions, but this may be beyond the scope of the current study.

Answer:

We would like to thank you for your valuable suggestion. We definitely agree that the question raised by our results can be summarized: how can focality actually impact the hormonal effects? Summarizing the challenges and providing a framework for further studies have been added to the discussion and conclusion section of our manuscript. Especially the cause, context and to what extend BC focality may impact hormonal effects would be of major interest for further studies. We have added this point in the conclusion section of our manuscript to encourage further studies. We also agree that this study does not aim to answer this question, nevertheless, should be subject of further examination.

Also, to this point, no standard international definition has been implemented for distinguishing multifocal and multicentric BC in detail. Several studies have previously described that the two entities were found to be predictive of a worse prognosis – nevertheless, if multifocal and multicentric BC can be regarded as equivalent in terms of aggressiveness of the disease should become subject of further studies. We have also added this important point to the discussion section.

See “Conclusion” (highlighted in manuscript document): “In summary, thyroid hormone modulated therapies should become subject of further investigations in regard to BC focality. Our results strengthen the need to further investigate the behavior of the nuclear receptor family in BC, especially in relation to focality. Further examinations studying the cause and to what extend BC focality may impact hormonal effects would be of major interest.“

See “Discussion” (highlighted in manuscript document):” “Distinguishing the BC entities may be regarded as the most important limitation of this study. Dividing BC entities into its subtypes may differ dependent on pathological centers and examiners. Especially for multifocal BC, defined by two or more separate tumor loci in the same quadrant, minimal distance between the separate tumors may result to be considered as unifocal BC. Thus, distinction between unifocal and multifocal BC may not always be clear. Also, to this point, no standard international definition has been implemented for distinguishing multifocal and multicentric BC [11, 12]. So far, several studies including Weissenbacher et al. have hypothesized that the two entities were found to be predictive of a worse prognosis. If multifocal and multicentric BC can be regarded as equivalent in terms of aggressiveness of the disease should become subject of further investigations.”

Reviewer 2 Report

Aim of this study was to assess the prognostic value of the steroid hormone receptor expression, counting the retinoid X receptor (RXR) and thyroid hormone receptors (THRs), on the two different breast cancer (BC) entities: multifocal/multicentric versus unifocal. It was studied in relation to overall survival and disease free survival and compared with histopathological grading and TNM status. The material consisted of 319 breast cancer patients. The expression of RXR and THR in its two isoforms, THRα1 and THRα2 was analyzed by immunohistochemistry and they showed significant correlation with both unifocal and multifocal breast cancer.

Introduction – Nicely describes breast cancer epidemiology and how focality are different aspects of the disease with different prognosis. Multifocal BC states two or more separate tumor loci in the same quadrant. Multicentric BC is understood as two or more separate invasive tumors in more than one quadrant of the same breast. Should this be considered equivalent in terms of aggressiveness of disease? Multifocal with minimal distance to the separate tumors may as well be considered unifocal. Probably not evaluated equally between different centra or between different pathologists. The transition between unifocal and multifocal may not always be clear. In this paper multifocality and multicentricity is considered as equal entities and compared to unifocal. I agree that this is the best way to separate and study focality but with the limitation mentioned above. This could be mentioned in the discussion part.

Material – well described

Results – well described with illustrative figures

RXR - with a p value of 0.036 a significant negative association of the DFS in multifocal/multicentric BC patients when expressing the RXR. Not significant in unifocal or on OS.

THRα1 - worse DFS for unifocal BC patients, when expressing the THRα1. This finding was confirmed by the Log-Rank test with a highly significant p value of 0.012. Regarding the OS of unifocal BC patients, THRα1 expression revealed no statistically significant difference (p=0.524). Using the same statistical devices for the multifocal and/or multicentric BC group, no significant correlations between prognosis and THRα1 expression could be outlined (DFS p=0.617; OS p=0.564).

THRα2 - Unifocal BC patients revealed no significant correlations between THRα2 expression and prognosis. The Kaplan-Meier survival analysis visualized, that patients with multifocal and/or multicentric BC have a better DFS, when expressing the THRα2.

Discussion – a review of the literature on this matter is adequate. The limitation mentioned above in respect to multifocal disease and how this entity may differ could be mentioned as a limitation to the study. The effect of these hormones are well described. It is still unclear why the hormones should have different effect on these different entities of breast cancer. How can focality have impact on the hormonal effects? The study is however not aimed to answer this question.

To conclude; Patients with multifocal/multicentric BC exposed a significant worse DFS when expressing RXR. THRα1 and THRα2 showed a prognostic association in both BC entities, but with major differences. Patients with unifocal BC showed a significant worse DFS when expressing THRα1. In contrast, a statistically significant positive association between THRα2 expression and enhanced DFS in multifocal/multicentric BC was shown. In summary, thyroid hormone modulated therapies should become subject of further investigations in regard to BC focality.

Author Response

Comments and Suggestions for Authors:

Aim of this study was to assess the prognostic value of the steroid hormone receptor expression, counting the retinoid X receptor (RXR) and thyroid hormone receptors (THRs), on the two different breast cancer (BC) entities: multifocal/multicentric versus unifocal. It was studied in relation to overall survival and disease free survival and compared with histopathological grading and TNM status. The material consisted of 319 breast cancer patients. The expression of RXR and THR in its two isoforms, THRα1 and THRα2 was analyzed by immunohistochemistry and they showed significant correlation with both unifocal and multifocal breast cancer.

Introduction – Nicely describes breast cancer epidemiology and how focality are different aspects of the disease with different prognosis. Multifocal BC states two or more separate tumor loci in the same quadrant. Multicentric BC is understood as two or more separate invasive tumors in more than one quadrant of the same breast. Should this be considered equivalent in terms of aggressiveness of disease? Multifocal with minimal distance to the separate tumors may as well be considered unifocal. Probably not evaluated equally between different centra or between different pathologists. The transition between unifocal and multifocal may not always be clear. In this paper multifocality and multicentricity is considered as equal entities and compared to unifocal. I agree that this is the best way to separate and study focality but with the limitation mentioned above. This could be mentioned in the discussion part.

Answer:

We would like to thank you for your valuable suggestion. We definitely agree and added this important limitation in the discussion section of our manuscript. We agree that distinguishing the BC entities may be regarded as the most important limitation of this study. Also, to this point, no standard international definition has been implemented for distinguishing multifocal and multicentric BC in detail. Several studies have previously described that the two entities were found to be predictive of a worse prognosis – nevertheless, if multifocal and multicentric BC can be regarded as equivalent in terms of aggressiveness of the disease should become subject of further studies. We have also added this important point to the discussion section.

See “Discussion” (highlighted in manuscript document):” “Distinguishing the BC entities may be regarded as the most important limitation of this study. Dividing BC entities into its subtypes may differ dependent on pathological centers and examiners. Especially for multifocal BC, defined by two or more separate tumor loci in the same quadrant, minimal distance between the separate tumors may result to be considered as unifocal BC. Thus, distinction between unifocal and multifocal BC may not always be clear. Also, to this point, no standard international definition has been implemented for distinguishing multifocal and multicentric BC [11, 12]. So far, several studies including Weissenbacher et al. have hypothesized that the two entities were found to be predictive of a worse prognosis. If multifocal and multicentric BC can be regarded as equivalent in terms of aggressiveness of the disease should become subject of further investigations.”

Material – well described

Results – well described with illustrative figures

RXR - with a p value of 0.036 a significant negative association of the DFS in multifocal/multicentric BC patients when expressing the RXR. Not significant in unifocal or on OS.

THRα1 - worse DFS for unifocal BC patients, when expressing the THRα1. This finding was confirmed by the Log-Rank test with a highly significant p value of 0.012. Regarding the OS of unifocal BC patients, THRα1 expression revealed no statistically significant difference (p=0.524). Using the same statistical devices for the multifocal and/or multicentric BC group, no significant correlations between prognosis and THRα1 expression could be outlined (DFS p=0.617; OS p=0.564).

THRα2 - Unifocal BC patients revealed no significant correlations between THRα2 expression and prognosis. The Kaplan-Meier survival analysis visualized, that patients with multifocal and/or multicentric BC have a better DFS, when expressing the THRα2.

Discussion – a review of the literature on this matter is adequate. The limitation mentioned above in respect to multifocal disease and how this entity may differ could be mentioned as a limitation to the study. The effect of these hormones are well described. It is still unclear why the hormones should have different effect on these different entities of breast cancer. How can focality have impact on the hormonal effects? The study is however not aimed to answer this question.

Answer:

We definitely agree that the question raised by our results is: how can focality actually impact the hormonal effects? Especially the cause, context and to what extend BC focality may impact hormonal effects would be of major interest. We have added this point in the conclusion section of our manuscript to encourage further studies. We also agree that this study does not aim to answer this question, nevertheless, should be subject of further examination.

See “Conclusion” (highlighted in manuscript document): “In summary, thyroid hormone modulated therapies should become subject of further investigations in regard to BC focality. Our results strengthen the need to further investigate the behavior of the nuclear receptor family in BC, especially in relation to focality. Further examinations studying the cause and to what extend BC focality may impact hormonal effects would be of major interest.“

To conclude; Patients with multifocal/multicentric BC exposed a significant worse DFS when expressing RXR. THRα1 and THRα2 showed a prognostic association in both BC entities, but with major differences. Patients with unifocal BC showed a significant worse DFS when expressing THRα1. In contrast, a statistically significant positive association between THRα2 expression and enhanced DFS in multifocal/multicentric BC was shown. In summary, thyroid hormone modulated therapies should become subject of further investigations in regard to BC focality.
